# The EAAT1 aspartate/glutamate transporter is dispensable for acute myeloid leukemia cell growth and response to therapy

Hernán A. Tirado[1], Nithya Balasundaram[1], Jean Jacobs[2], Fleur Leguay[1], Lotfi Laaouimir[1], Nick van Gastel[1,3]*

1 Cellular Metabolism and Microenvironment Laboratory, de Duve Institute, UCLouvain, Brussels, Belgium, 2 Biochemistry and Metabolic Research Group, de Duve Institute, UCLouvain, Brussels, Belgium, 3 WELBIO Department, WEL Research Institute, Wavre, Belgium

* nick.vangastel@uclouvain.be

## Abstract

Acute myeloid leukemia (AML) is an aggressive malignancy of hematopoietic stem and progenitor cells characterized by profound metabolic dysregulation. Pyrimidine biosynthesis has emerged as a critical metabolic dependency in AML, but clinical translation has been hampered by unacceptable toxicity of current pyrimidine synthesis inhibitors. Since aspartate is an essential nutrient for pyrimidine biosynthesis, we investigated the role of aspartate import via the excitatory amino acid transporter 1 (EAAT1) in AML. We found that EAAT1 is broadly expressed across AML cell lines and patient samples, with enrichment in M4 and M5 subtypes and increasing levels following chemotherapy treatment. Pharmacological inhibition of EAAT1 impaired AML cell viability *in vitro*, but metabolomic profiling and nutrient rescue experiments showed that these effects were independent of intracellular aspartate levels. Moreover, AML cells cultured in aspartate-free medium maintained proliferation and did not become more sensitive to chemotherapy. EAAT1 inhibition in mice increased bone marrow plasma aspartate levels, confirming inhibition of cellular aspartate uptake, but did not affect growth or chemosensitivity of MLL-AF9-expressing AML cells *in vivo*. These findings suggest that AML cells possess several complementary mechanisms to support their aspartate requirements and that EAAT1 inhibition does not impair AML growth or response to chemotherapy.

## Introduction

Acute myeloid leukemia (AML) is a genetically and phenotypically heterogeneous disease that arises and evolves within the bone marrow microenvironment, where both normal and malignant hematopoiesis coexist and interact [1,2]. For decades, standard AML therapy has centered on the use of cytotoxic agents such as nucleoside analogs, particularly cytarabine, which target rapidly proliferating cells by interfering

**Data availability statement:** All relevant data are within the manuscript and its Supporting information files.

**Funding:** This work was supported by the Belgian Foundation Against Cancer [F/2020/1440, F/2024/2556], the Fund for Scientific Research - FNRS [F.R.S.-FNRS; A5/5-CQ/135, M4/1/2/5-MIS/BEJ] and the de Duve Institute. H.A.T. and J.J. were supported by doctoral research fellowships from the Fund for Research Training in Industry and Agriculture (FRIA) [FRIA/FC-2577 and FRIA/FC-2574] and by a Bourse du Patrimoine from the UCLouvain. L.L. was supported by a FRIA doctoral research fellowship [FRIA/FC-4456], N.B. was supported by a post-doctoral Chargée de Recherches fellowship from the F.R.S.-FNRS [A4/5-MCF/DEA-CR 27]. The funders had no role in study design, data collection and analysis, decision to publish, or preparation of the manuscript.

**Competing interests:** The authors have declared that no competing interests exist.

with DNA replication, often in combination with anthracyclines. Over the past years, new chemotherapy regimens adapted for older patients and targeted oral therapies against driver mutations have been added to the repertoire [3]. However, resistance to therapy remains a major clinical hurdle, contributing to poor long-term survival outcomes in most patients [4]. While significant efforts have focused on deciphering the genetic and cytokine-mediated mechanisms of resistance, recent advances underscore the critical role of cellular metabolism and metabolic crosstalk within the bone marrow niche in shaping leukemic progression and therapeutic relapse [5].

Altered metabolic programs are increasingly recognized as a hallmark of AML, contributing to its uncontrolled proliferation, impaired differentiation, and therapy resistance [6–8]. A well-characterized example involves mutations in isocitrate dehydrogenase (IDH), which result in the production of the oncometabolite R-2-hydroxyglutarate from α-ketoglutarate (αKG). This metabolite inhibits αKG-dependent dioxygenases, thereby promoting leukemogenesis [9]. Although targeted therapies for IDH-mutant AML are now clinically approved, targeting the metabolic vulnerabilities of AML subtypes lacking mutations in metabolic enzymes has proven difficult. For example, AML cells frequently display changes in their energy metabolism, including glycolysis, the tricarboxylic acid (TCA) cycle, and oxidative phosphorylation (OXPHOS) [7,8,10]. Notably, OXPHOS activity is closely linked to the maintenance of leukemia stem cells, a population that drives relapse [11,12]. Unfortunately, even the newest generation of OXPHOS-targeting drugs such as IACS-010759 have shown poor clinical efficacy and significant toxicity [13]. Nucleotide biosynthesis has also emerged as a key metabolic dependency in AML, with the inhibition of *de novo* pyrimidine or purine biosynthesis shown to induce differentiation and cell death in preclinical models [14–16]. However, clinical translation has again been challenging, with concerns about the toxicity of existing nucleotide synthesis inhibitors [17,18].

The limitations of existing metabolism-targeting drugs highlight the need to identify alternative strategies for selectively exploiting the metabolic vulnerabilities of AML cells. In our recent work, we have identified aspartate as one of the metabolites transferred from bone marrow stromal cells to leukemic cells, where it contributes to pyrimidine biosynthesis [19]. Aspartate, a non-essential amino acid ubiquitously synthesized by mammalian cells, plays a central role not only in nucleotide synthesis, but also contributes to protein biosynthesis, redox balance, TCA cycle anaplerosis, nitrogen shuttling, and mitochondrial metabolism [20,21]. Remarkably, we found that aspartate concentrations in the bone marrow are 70- to 100-fold higher than in peripheral blood, suggesting a unique metabolic landscape supporting leukemic metabolism [19]. Further supporting this, studies across multiple cancer types have implicated aspartate as a limiting metabolite under hypoxic conditions [22]. Given the physiologically low oxygen tension of the bone marrow niche [23], AML cells may be particularly reliant on exogenous aspartate when growing in this environment.

Based on the established dependency of AML on pyrimidine biosynthesis and the notion that the metabolic constraints of the bone marrow microenvironment may limit aspartate synthesis, we hypothesized that targeting aspartate uptake could serve as an upstream strategy to disrupt pyrimidine metabolism more selectively and safely.

Excitatory amino acid transporters (EAATs) have traditionally been studied for their role in glutamate clearance in the central nervous system, but they also transport aspartate and are increasingly recognized as key regulators of amino acid metabolism in non-neuronal tissues, including cancer [24,25]. In the current study, we investigated whether AML cells express and rely on EAATs for aspartate uptake, and whether inhibition of these transporters impacts AML cell growth or response to therapy.

## Methods

### Ethics statement

All animal procedures performed in this study were approved by the UCLouvain Institutional Animal Care and Use Committee (approval number 2021/UCL/MD/03). For bioluminescence imaging, mice were anesthetized through isoflurane inhalation (2.5% in air for sleep induction and 1.5% in air during imaging). At the end of the experiments, mice were euthanized by $CO_2$ inhalation.

Immortalized human cell lines were commercially obtained and their use for scientific research is exempt from ethical review in accordance with Belgian law. No new human material was collected in this study. Publicly available human gene and protein expression data were analyzed anonymously.

### Public datasets of gene and protein expression

We obtained values of gene expression for *SLC1A1, SLC1A2* and *SLC1A3* from BloodSpot [26] (www.bloodspot.eu, datasets: BeatAML, MILE) and Vizome (www.vizome.org, datasets: BeatAML2.0, attributes: FAB BlastMorphology, isDenovo, isTherapy). Protein levels of EAAT1, EAAT2 and EAAT3 in AML were obtained from The Cancer Genome Atlas (TCGA) [27] (proteomics.leylab.org, dataset: TMT).

### Cell lines

MA9–1 cells were obtained by retroviral infection of whole bone marrow cells with the humanized MLL-AF9 fusion oncogene as previously described [28]. MA9–2 cells were obtained after crossing MLL-AF9 knock-in mice with ubiquitous GFP-expressing mice, as well as luciferase-expressing mice, and have been previously described [19]. Both cell models were cultured at 37°C in a humidified incubator with ambient air and 5% $CO_2$ in RPMI1640 medium (ThermoFisher, 31870074) supplemented with 10% fetal bovine serum (FBS; Sigma Aldrich, F7524), 100 I.U./mL penicillin-100 µg/mL streptomycin (ThermoFisher, 15140122), 2 mM L-glutamine (Fisher Scientific, 11510626), 20 ng/mL recombinant mouse stem cell factor, 10 ng/mL recombinant mouse interleukin 3 and 10 ng/mL recombinant mouse interleukin 6 (BioLegend Europe BV, 579708, 575708, 575508). U937, NB4, MV4–11, MOLM14, OCI-AML3, NOMO-1 and MONOMAC6 (MM6) human AML cell lines were obtained from the Leibniz Institute DSMZ and cultured at 37°C in a humidified incubator with ambient air and 5% $CO_2$ in RPMI1640 medium supplemented with 10% FBS, 100 I.U./mL penicillin-100 µg/mL streptomycin and 2 mM L-glutamine. Cell line identity was validated by DNA fingerprinting and cultures were tested for mycoplasma contamination every six months using a Mycoplasma PCR Detection Kit (Abcam, ab289834). For aspartate depletion experiments, custom-made aspartate-free, glutamine-free RPMI1640 medium (ThermoFisher) was supplemented with 10% dialyzed FBS (Fisher Scientific, 11520646), 100 I.U./mL penicillin-100 µg/mL streptomycin and 2 mM L-glutamine.

### Real-time quantitative PCR (RT-qPCR) analysis

Total RNA was extracted from the cells using TRIzol RNA extraction reagent (Fisher Scientific, 12044977) and cDNA was synthesized from 1 µg of RNA using the RevertAid First Strand cDNA Synthesis Kit (ThermoFisher, K1622). RT-qPCR was performed on a QuantStudio 3 Real Time PCR system (ThermoFisher). For human cells, the PowerTrack SYBR Green Master Mix (ThermoFisher, A46109) was used with the following primers: h*SLC1A3*-Forward: 5'- TCT TGC CAC

TCC TCT ACT TC −3'; h*SLC1A3*-Reverse: 5'- TTG TCC ACG CCA TTG TTC −3'; h*ACTB*-Forward: 5'- AGC GAG CAT CCC CCA AAG TT −3'; h*ACTB*-Reverse: 5'- AGG GCA CGA AGG CTC ATC ATT −3'. For mouse cells, the TaqMan Fast Advanced Master Mix (ThermoFisher, 4444964) was used with the following pre-designed PrimeTime Std qPCR Assays with 6-FAM/ZEN/IBFQ reporter/quencher (Integrated DNA Technologies): m*Slc1a1* (Mm.PT.58.17503262), m*Slc1a2* (Mm.PT.58.41489267), m*Slc1a3* (Mm.PT.58.5217423), m*Actb* (Mm.PT.58.33540333). Assays were performed in technical duplicate, and data were analyzed by the $2^{-\Delta Ct}$ method, relative to β-actin (*ACTB*).

## Chemicals

TFB-TBOA (Axon MedChem, 2640), UCPH-101 (Bio-Techne, 3490/50), dimethyl-aspartate (Sigma Aldrich, 456233-5G), dimethyl-alpha-ketoglutarate (Sigma Aldrich, 349631-5G), uridine (Sigma Aldrich, U3750-25G) and tyrosine (Sigma Aldrich, T8566) were added to cell cultures at the indicated concentrations. For *in vitro* chemotherapy treatment, doxorubicin (Sigma Aldrich, C6645-1G) at 30 nM and cytarabine (Sigma Aldrich, D1515-10MG) at 10 nM or 1 µM were added to cultures simultaneously.

## MTT assay

Cells were incubated with indicated drugs/molecules, then treated with 1.5 mg/mL Thiazolyl Blue Tetrazolium Bromide (MTT, Fisher Scientific, 15234654) for 4 hours, followed by cell lysis with 10% SDS (Sigma Aldrich, 05030–500ML-F) and overnight incubation. The absorbance was read at 560 nm with a reference wavelength of 630 nm using a Glomax (Promega). Half-maximal inhibitory concentrations (IC50) were calculated using GraphPad Prism 10.

## Cell death assay

Cells were treated with the indicated drugs or molecules for 24 hours and then harvested and stained with annexin-V-APC (BioLegend, 640941) and Propidium Iodide (PI; Sigma Aldrich, P4864-10ML) according to the manufacturer's protocol. Samples were analyzed by flow cytometry.

## Cell cycle assay

Cells were treated with indicated molecules for 6 hours, harvested and washed with PBS. The pellet was fixed with 2 mL of ice-cold 70% ethanol (Avantor, 20.821.310) in a drop-wise manner while gently vortexing, kept for 30 minutes on ice and centrifuged at 1000 x g for 5 minutes. Cells were then washed twice with PBS and incubated for 20 minutes with 200 µL of 1 µg/mL DAPI (Roche, 10236276001). Cells were washed and analyzed by flow cytometry.

## Metabolomics analysis

For metabolomics analysis by gas chromatography-coupled mass spectrometry (GC-MS), cells were treated for 6 hours with indicated molecules, harvested and washed twice in saline (NaCl 0,9%, Sigma Aldrich, 71376−1 KG). For aspartate uptake analysis, cells were cultured for 6 hours in aspartate-free RPMI1640 medium supplemented with 150 µM U-$^{13}C_4$-aspartate (Cambridge Isotope Laboratories, CLM-1801-H). The cell pellets were resuspended and vortexed thoroughly in 0.5 mL of methanol (kept at −20°C, Sigma Aldrich, 900688-1L), after which 0.5 mL ice-cold water and 1 mL of chloroform (Sigma Aldrich, 650471-1L) were added and samples were agitated for 15 minutes at 4°C. Samples were then centrifugated at 14,000 x g for 10 minutes at 4°C, after which the polar (top) phase was collected and dried in a Speed-Vac vacuum concentrator calibrated at −100°C. Metabolites were resuspended in 15 µL of methoxamine (Fisher scientific, 11567630) and incubated for 90 minutes at 30°C with shaking at 300 rpm. Next, 30 µL of MSTFA (Macherey-Nagel, MN701270.201) was added, samples were vortexed for 5 seconds and incubated for 30 minutes at 37°C. The mix was then centrifugated again at 14,000 x g for 10 minutes and 40 µL were transferred into a GC-MS glass vial. Samples were

analyzed by GC-MS (ThermoScientific, Trace 1310, Agilent CP-Sil 8 CB-MS 30 m x 0.25 mm x 0.25 µm capillary column) on the same day of derivatization.

### Generation of human *SLC1A3*-knockout U937 cells by CRISPR/Cas9

Cells were transfected using the Nucleofector 4D-X system (Lonza) according to the manufacturer's instructions. Briefly, 10,000 U937 cells were resuspended in 20 µL of Nucleofector Solution (P3 kit; Lonza, V4XP-3032) and mixed with a duplex crRNA:tracrRNA complex and TrueCut Cas9 Protein v2 (ThermoFisher, A36498). The Alt-R CRISPR-Cas9 tracrRNA (Integrated DNA Technologies, 1072533) was used and the guide RNAs against *SLC1A3* were: 5'- AGC ACC CAC AAG CGT TTC GT −3' (crRNA1), 5'- AGC TCA TTC TGT ATG GTC GG −3' (crRNA2), and a negative control crRNA (Integrated DNA Technologies). The mixture was nucleofected using program EO100 and cells were immediately transferred to prewarmed complete growth medium and allowed to recover. Clonal lines were then generated by plating the nucleofected cells at limiting dilution to obtain a single cell per well. The resulting clones were expanded and subjected to RT-qPCR to assess knockout efficiency.

### Mouse model and drug treatments

8- to 10-week-old male and female C57Bl/6 mice were transplanted with 1 million murine MLL-AF9-GFP-luciferase cells (MA9–2) through intravenous injection (tail vein). Leukemia engraftment and disease progression were monitored using bioluminescence imaging on an IVIS In Vivo Imaging System (PerkinElmer) after injecting mice with D-Luciferin (ZellBio Gmbh, LUCK-1G), as previously described [19]. For single agent EAAT inhibitor treatments, animals were randomized 4 days after cell injection and treated daily with UCPH-101 (20 mg/kg in 10% DMSO, 40% PEG300, 5% Tween-80, 45% saline) or TFB-TBOA (20 mg/kg in saline) through intraperitoneal injection until day 21. For combination with chemotherapy, treatment with UCPH-101 or TFB-TBOA was started at day 7 and continued for 2 weeks. Treatment with standard induction chemotherapy (iCT) started at day 9 and consisted of cytarabine (100 mg/kg in saline) given once daily for 5 days and doxorubicin (3 mg/kg in saline) given once daily for the first 3 days, via intraperitoneal injection.

For survival experiments, mice were monitored daily from day 0 to day 15 and twice per day starting on day 15 after AML cell engraftment. Mice were euthanized by $CO_2$ asphyxiation within 2 hours after reaching humane endpoints, which were defined using scoring sheets monitoring overall body condition, physical appearance (fur, posture, eyes) and animal behavior (reactivity, mobility). No specific analgesics, anesthetics or other welfare measures were used. Experiments were designed using 5 mice per group and continued until the last mouse reached the humane endpoints and was euthanized; no animals died before meeting the criteria for euthanasia.

### Statistical analysis

All results are reported as mean±standard deviation (SD). Statistical significance of the difference between experimental groups was analyzed by two-tailed unpaired Student's t-test, one-way ANOVA with Bonferroni post-hoc test or the logrank (Mantel-Cox) test for the Kaplan-Meier survival curve analyses using the GraphPad Prism 10 software. Differences were considered statistically significant at $P < 0.05$.

## Results

### AML cells express *SLC1A3*

Aspartate can be transported across the plasma membrane by members of the EAAT family, and particularly by EAAT1 (encoded by *SLC1A3*), EAAT2 (encoded by *SLC1A2*) and EAAT3 (encoded by *SLC1A1*) [25]. We analyzed the expression of these putative aspartate transporters in human AML cells using existing databases containing gene expression

data of AML patient samples, including those with different molecular subtypes. In the BEAT-AML cohort [29], we observed that *SLC1A3* is highly expressed in AML cells across molecular subtypes compared to *SLC1A2* and *SLC1A1* (Figs 1A, S1A and S1B). Further investigation based on the FAB classification revealed that myelomonocytic (AML-M4) and monocytic (AML-M5) AML subtypes had the highest expression of *SLC1A3* (Fig 1B). We also noted that relapsed AML had higher expression of *SLC1A3* when compared to diagnostic specimens (Fig 1C). Examination of the data of the MILE study [30] confirmed higher expression of *SLC1A3* compared to *SLC1A2* and *SLC1A1* in AML cells, and showed that myeloid blood cancers, including AML, express slightly higher levels of *SLC1A3* compared to lymphoid leukemias (S1C–S1E Fig). The gene expression results were confirmed at the protein level using proteomics data from The Cancer

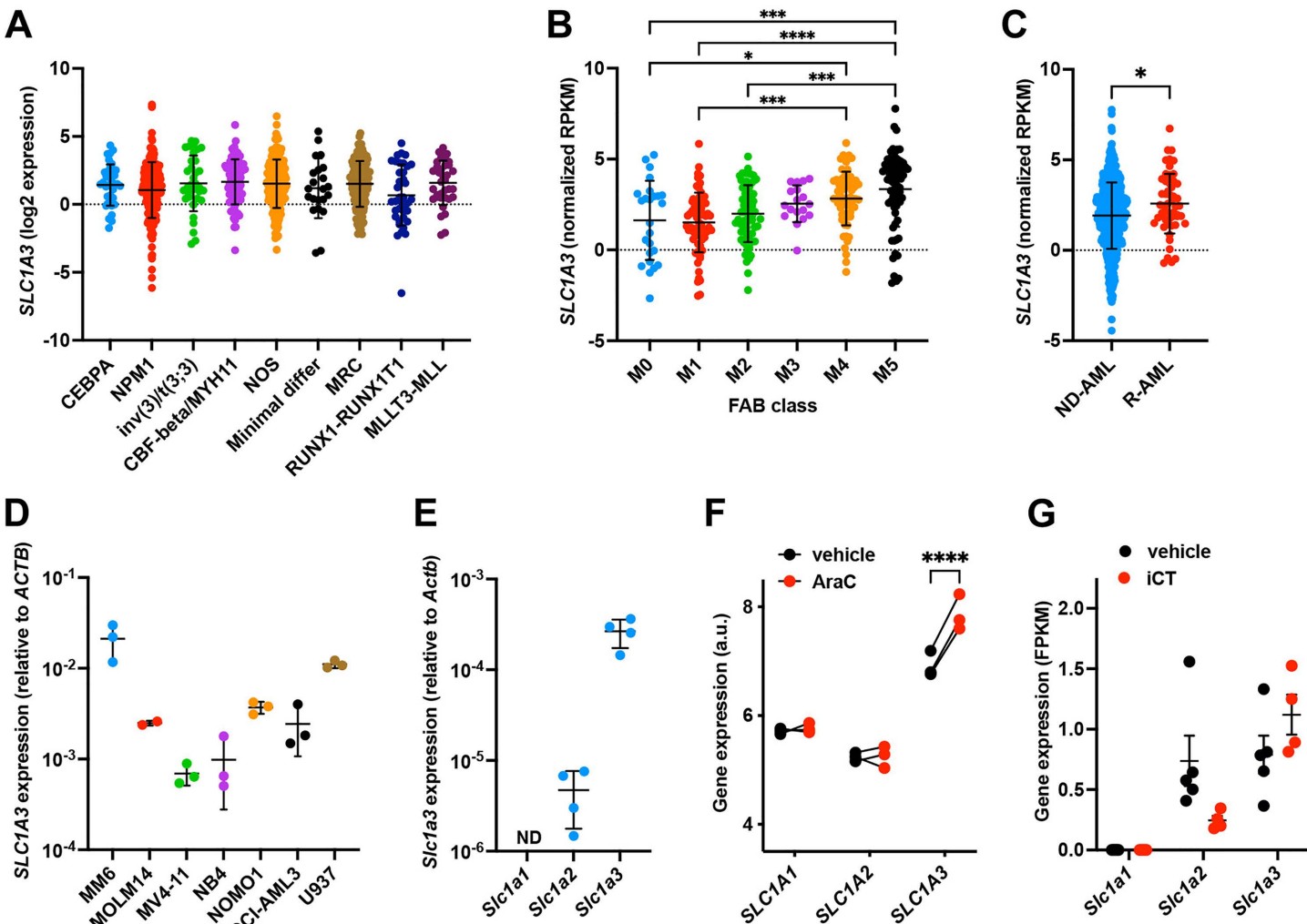

**Fig 1. AML cells express *SLC1A3*. A-C.** Expression of *SLC1A3* in AML cells across different patient genetic subgroups (A), according to FAB classification (B) and comparing newly diagnosed (ND) and relapsed (R) patients (C) in the BEAT-AML cohort. **D-E.** Expression of *SLC1A3* in human AML cell lines (D) and of *Slc1a1*, *Slc1a2* and *Slc1a3* in a mouse AML MLL-AF9 cell line (E) as quantified by qRT-PCR and expressed relative to β-actin (*ACTB*). **F.** Expression of *SLC1A1*, *SLC1A2* and *SLC1A3* by human AML patient-derived xenograft cells isolated from mice treated with cytarabine or vehicle. Data obtained from (31). **G.** Expression of *Slc1a1*, *Slc1a2* and *Slc1a3* by mouse AML cells (MA9−2 cell line) isolated from mice treated with induction chemotherapy (iCT; doxorubicin 3 mg/kg and cytarabine 100 mg/kg given in a 5 + 3 regimen) or vehicle. Data obtained from (19). Data are presented as mean ± SD. *P < 0.05; ***P < 0.001; ****P < 0.0001.

Genome Atlas (TCGA [27]), which showed the presence of EAAT1, but not EAAT2 or EAAT3, in AML patient cells, with the highest levels found in M4 and M5 subtypes (S1F Fig).

We confirmed the expression of *SLC1A3* across different human AML cell lines and observed that the monocytic MM6 and U937 cell lines had the highest expression compared to other AML cell lines (Fig 1D). Mouse MLL-AF9-expressing AML cells showed lower expression of *Slc1a3* compared to the human cell lines, but *Slc1a3* was still expressed at much higher levels compared to *Slc1a2*, while *Slc1a1* expression was undetectable (Fig 1E).

To better understand the effects of therapy on *SLC1A3* expression, we re-analyzed previously published RNA sequencing datasets of human and mouse AML cells obtained from engrafted mice treated with induction chemotherapy (iCT; doxorubicin + cytarabine) or cytarabine (AraC) alone [19,31], which showed that both human and mouse residual AML cells exhibit increased expression of *SLC1A3* after therapy (Fig 1F and 1G).

## EAAT1 inhibitors reduce AML growth *in vitro*

To investigate the importance of exogenous aspartate for AML cell growth, we exposed both mouse and human cell lines to known EAAT1 inhibitors, and assessed the effects on cell health, apoptosis and proliferation. We first used increasing concentrations of TFB-TBOA (an aspartate analog and competitive inhibitor of EAAT1, EAAT2, and EAAT3 [32]) and UCPH-101 (an uncompetitive allosteric inhibitor selective for EAAT1 [33]) and measured overall cell health of MA9–1 and U937 cells using the MTT assay. Metabolic activity of both cell lines was reduced in a dose-dependent manner by the aspartate transporter inhibitors; however, UCPH-101 was found to be more potent (Fig 2A). We repeated this experiment with a larger number of human and mouse AML cell lines and calculated the half-maximal inhibitory concentration (IC50) for TFB-TBOA and UCPH-101. All tested cell lines (MM6, MOLM14, MV4–11, NB4, NOMO1, OCI-AML3 and U937 human AML cell lines, MA9–1 and MA9–2 mouse AML cell lines) responded similarly to both molecules, with UCPH-101 demonstrating consistently higher potency (Fig 2B).

Given that the MTT assay measures overall metabolic health, we further explored how the inhibitors affected AML cells by measuring apoptosis and performing cell cycle analysis. In both the mouse MA9–1 and human U937 cell line we observed increased apoptosis when cells were exposed to 20 µM of UCPH-101 for 24 hours, whereas 250 µM of TFB-TBOA did not affect cell viability (Fig 2C). Cell cycle analysis, conducted after 6 hours of UCPH-101 treatment (20 µM) to prevent cell death, revealed fewer cells in the G1 phase and a corresponding increase in cells in the S and G2/M phases, indicating a dysregulation of the cell cycle (Fig 2D). Taken together, these results show that AML cells are sensitive to aspartate transporter inhibitors, with UCPH-101 showing the highest potency.

## UCPH-101 kills AML cells in an aspartate-independent manner

To evaluate the metabolic effects of UCPH-101, we treated MA9−1, MA9−2, and U937 cell lines with this inhibitor and quantified intracellular metabolites by GC-MS. After 6 hours of incubation, aspartate levels were reduced, along with tyrosine and several TCA cycle metabolites (citrate, isocitrate, alpha-ketoglutarate, fumarate, malate). In contrast, essential amino acid levels were increased (isoleucine, leucine, lysine, methionine, phenylalanine, valine), demonstrating a significant metabolic effect of the inhibitor (Fig 3A and 3B).

We hypothesized that blockade of aspartate uptake was responsible for the observed metabolic changes, since aspartate can directly feed the TCA cycle or have indirect effects on mitochondrial metabolism through its role in nucleotide synthesis, while the increase in essential amino acid levels may be linked to a lower proliferation or overall decrease in translation when cells are deprived of aspartate. To counteract the effects of the inhibitor, we treated the AML cell lines with both UCPH-101 and dimethyl-aspartate (dm-Asp), a membrane-permeable form of aspartate. GC-MS analysis confirmed rescue of intracellular aspartate levels by dm-Asp in mouse and human AML cell lines treated with UCPH-101, while the observed changes in the levels of TCA cycle metabolites or essential amino acids were not restored (Fig 3C and 3D). However, addition of dm-Asp did not rescue cell viability, suggesting that UCPH-101 induces AML cell death and

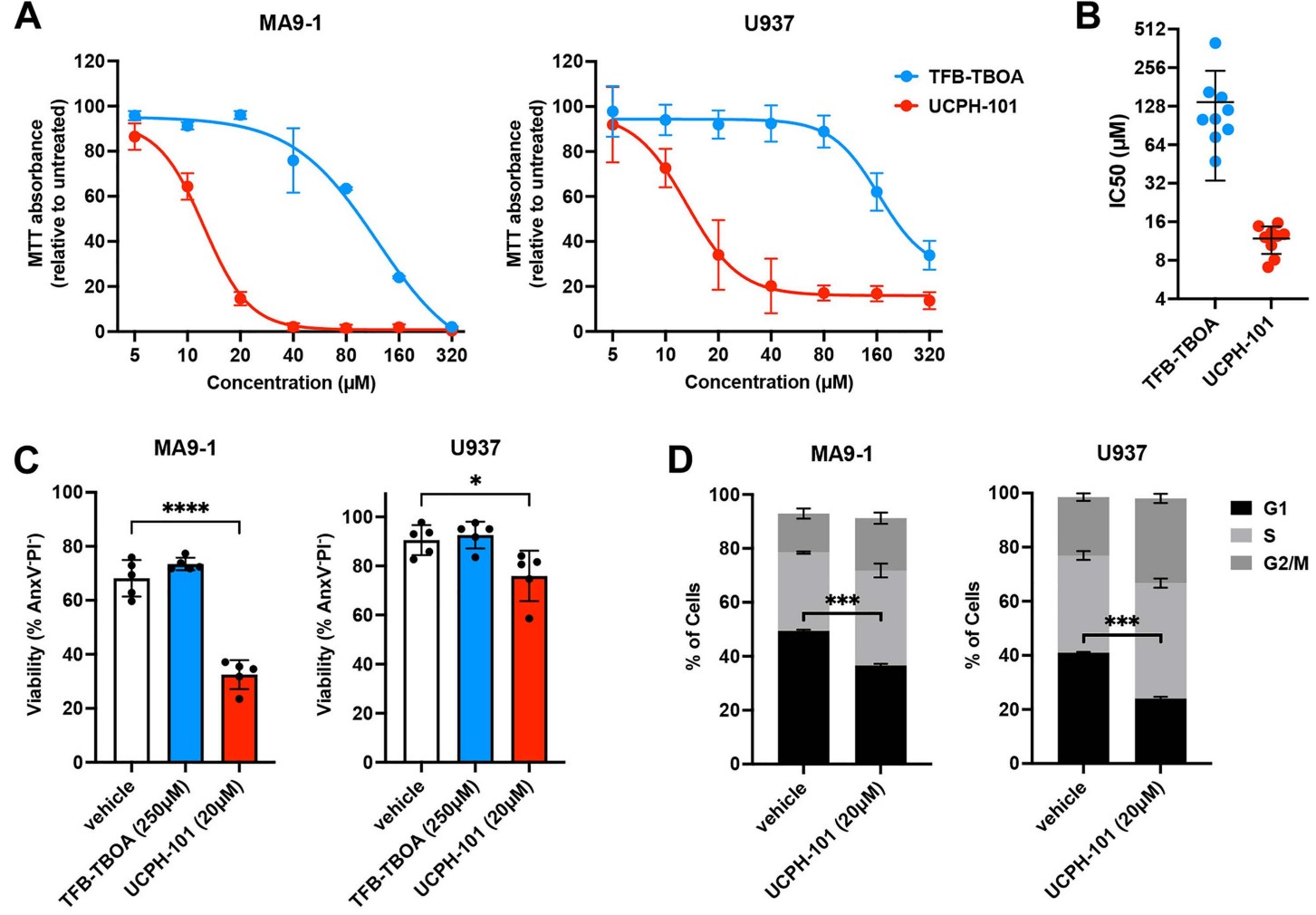

**Fig 2. EAAT1 inhibitors reduce AML growth *in vitro*. A.** Overall health of AML cells exposed to increasing concentrations of TFB-TBOA or UCPH-101 for 24 hours, as determined by the MTT assay. **B.** Half-maximal inhibitory concentration (IC50) for TFB-TBOA and UCPH-101 across 9 different human and mouse AML cell lines, based on the MTT assay. Each dot represents a different AML cell line. **C.** Viability of AML cells treated with TFB-TBOA (250 μM) or UCPH-101 (20 μM) for 24 hours as measured by flow cytometry. **D.** Cell cycle analysis of AML cells treated with UCPH-101 (20 μM) for 6 hours as measured by flow cytometry. Data are presented as mean±SD. *P < 0.05; ***P < 0.001; ****P < 0.0001.

causes metabolic changes through a mechanism unrelated to intracellular aspartate levels (Fig 3E). Based on the other metabolic effects observed after treating AML cells with UCPH-101 (Fig 3A and 3D), we supplemented the medium with dimethyl-alpha-ketoglutarate (dm-aKG), uridine, or tyrosine in the presence of UCPH-101. We found that none of these supplements restored cell viability (Fig 3F–3H). These data show that when used at 20 μM, UCPH-101 kills AML cells through a mechanism independent of aspartate uptake, suggesting that either EAAT1 has other substrates, or UCPH-101 has other targets.

## AML cells do not require extracellular aspartate *in vitro*

To better understand the importance of extracellular aspartate for AML cells and the effects of the EAAT1 inhibitors, we performed an uptake assay using $^{13}C_4$-labeled aspartate. Both MA9−2 and U937 AML cells took up extracellular aspartate,

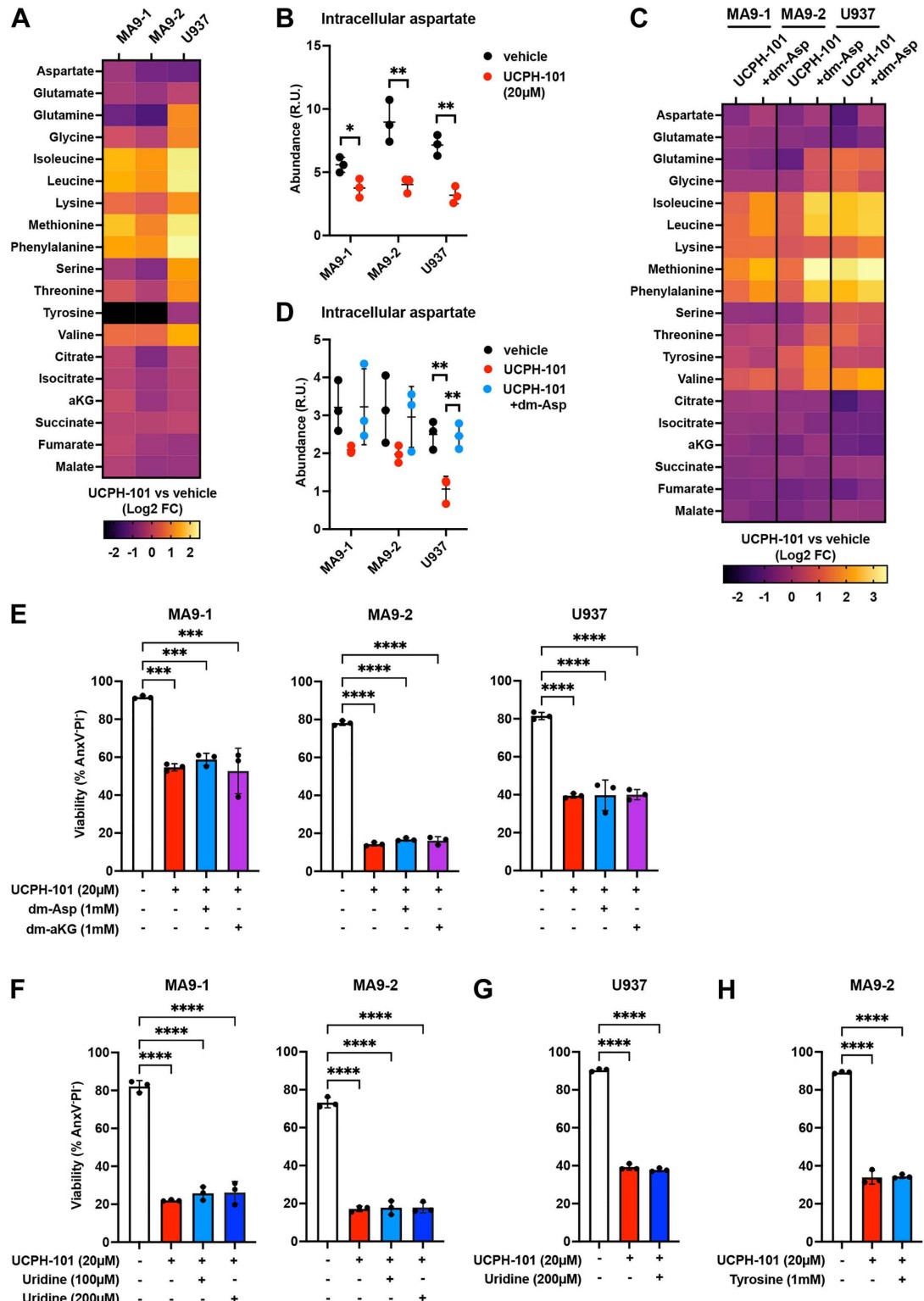

**Fig 3. UCPH-101 kills AML cells in an aspartate-independent manner. A.** Heatmap showing relative intracellular levels of different metabolites in AML cells treated with UCPH-101 (20 μM) for 6 hours as measured by GC-MS. aKG: alpha-ketoglutarate. **B.** Quantification of intracellular levels of aspartate in AML cells treated with UCPH-101 (20 μM) for 6 hours, as measured by GC-MS (representation of aspartate data from panel A). R.U.:

Relative Units. **C.** Heatmap showing relative intracellular levels of different metabolites in AML cells treated with UCPH-101 (20 µM) with or without dimethyl-aspartate (dm-Asp; 1 mM) for 6 hours as measured by GC-MS. **D.** Quantification of intracellular levels of aspartate in AML cells treated with UCPH-101 (20 µM) with or without dimethyl-aspartate (dm-Asp; 1 mM) for 6 hours, as measured by GC-MS (representation of aspartate data from panel C). **E.** Viability of AML cells treated with UCPH-101 with or without dimethyl-aspartate (dm-Asp) or dimethyl-alpha-ketoglutarate (dm-aKG) at the indicated concentrations for 24 hours, as measured by flow cytometry. **F-G.** Viability of mouse MLL-AF9 AML cells (F) or human U937 cells (G) treated with UCPH-101 with or without uridine at the indicated concentrations for 24 hours, as measured by flow cytometry. **H.** Viability of mouse MLL-AF9 AML cells treated with UCPH-101 with or without tyrosine at the indicated concentrations for 24 hours, as measured by flow cytometry. Data are presented as mean±SD. **P<0.01; ***P<0.001; ****P<0.0001.

with 10−15% of the total cellular aspartate pool labeled after 6 hours of incubation (Fig 4A). Surprisingly, UCPH-101 did not have any effects on aspartate uptake, even though it reduced intracellular aspartate levels as previously observed, again underscoring that the induction of AML cell death by this compound is unrelated to aspartate uptake. In contrast, TFB-TBOA at 250 µM significantly reduced aspartate uptake by both MA9−2 and U937 cells and moderately decreased total intracellular aspartate levels (Fig 4B), showing that this molecule is an effective inhibitor of aspartate transport into AML cells.

Given that TFB-TBOA is a competitive inhibitor that may block other aspartate transporters besides EAAT1, we next generated *SLC1A3*-knockout U937 cells through CRISPR/Cas9 to specifically study the role of this transporter (S2A Fig). Loss of *SLC1A3* did not reduce $^{13}C_4$-aspartate uptake by U937 cells or overall aspartate pools (Fig 4C) and accordingly had no effects on overall AML cell viability or cell cycle progression (Figs 4D and S2B). Thus, while AML cells take up extracellular aspartate in culture, they do not depend on EAAT1 to do so. U937 cells lacking *SLC1A3* also showed the same response to UCPH-101 (S2C Fig), further confirming that this inhibitor exerts its cytotoxic effects through EAAT1-independent mechanisms.

To better understand the importance of aspartate as a nutrient for AML cells, we next cultured MA9–1 and MA9–2 mouse AML cell lines in regular RPMI-1640 medium (containing 150 µM aspartate), aspartate-free medium, and medium containing 1 mM aspartate (to mimic aspartate levels in the bone marrow plasma [19]). Given that electron transport chain activity is essential for *de novo* aspartate synthesis [34,35] and hypoxia has been shown to limit the ability of cancer cells to synthetize aspartate [22], we further tested whether lowering oxygen tension (21%, 2% or 1% $O_2$) would change the reliance of AML cells on extracellular aspartate. Population doubling analysis revealed that AML cells grew normally under all conditions, regardless of extracellular aspartate levels (Fig 4E). Expectedly, lower oxygen availability decreased cell growth, but it did not increase sensitivity to extracellular aspartate depletion. These findings show that AML cells do not require exogenous aspartate for *in vitro* growth.

Since exposure to iCT increased EAAT1 levels in AML cells (Fig 1H), we wondered if the stress of the therapy would generate a higher dependency on EAAT1 or extracellular aspartate in AML cells. We exposed AML cells to iCT in combination with TFB-TBOA, knockout of *SLC1A3* or with aspartate depletion from the medium. However, none of the strategies to block aspartate uptake sensitized AML cells to iCT (Fig 4F–4H), indicating that EAAT1 or exogenous aspartate is also not required for AML cells to survive the consequences of chemotherapy *in vitro*.

## Inhibition of EAAT1 does not alter AML growth or chemotherapy response *in vivo*

While AML cells do not appear to require extracellular aspartate when cultured *in vitro*, we hypothesized that in the bone marrow microenvironment these cells may depend on aspartate as a nutrient for their growth or survival following iCT treatment. Since direct depletion of aspartate in the bone marrow plasma would be challenging, we tested whether treatment of mice with TFB-TBOA or UCPH-101 could impact aspartate uptake by cells in the bone marrow. To this end, we treated control C57Bl/6 mice daily with TFB-TBOA (for 4 or 8 days) or UCPH-101 (for 2 days) and isolated the bone marrow plasma. Analysis by GC-MS revealed elevated levels of aspartate in the bone marrow plasma after treatment of mice with the EAAT1 inhibitors, suggesting an accumulation of the metabolite due to reduced cellular consumption and

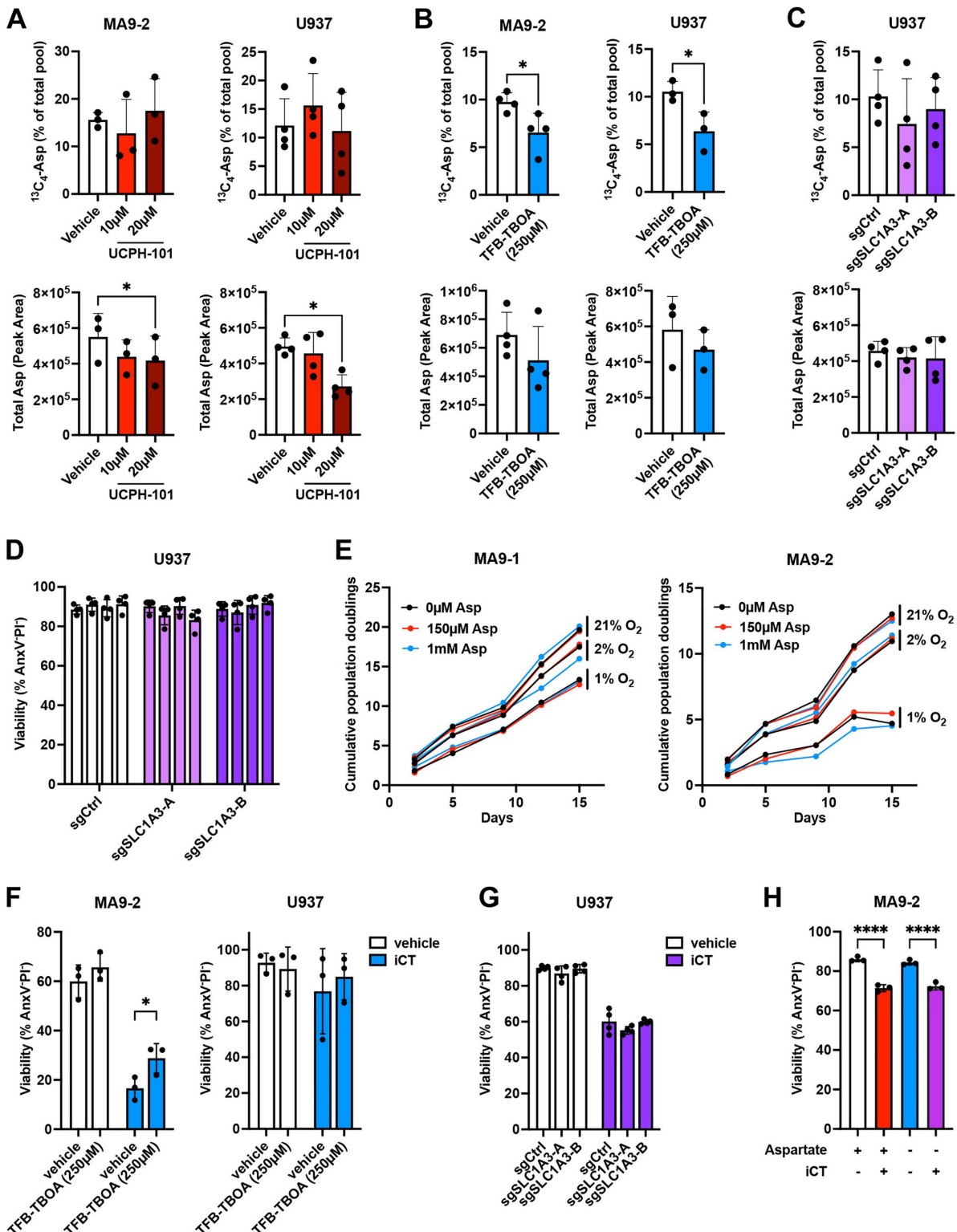

**Fig 4. AML cells do not require extracellular aspartate *in vitro*. A.** Effect of UCPH-101 on the uptake of $^{13}C_4$-aspartate (*top*) and on total intracellular aspartate levels (*bottom*) in MA9−2 and U937 AML cells after 6 hours of incubation. **B.** Effect of TFB-TBOA on the uptake of $^{13}C_4$-aspartate (*top*) and on total intracellular aspartate levels (*bottom*) in MA9−2 and U937 AML cells after 6 hours of incubation. **C.** Effect of *SLC1A3* knockout on the uptake

of $^{13}C_4$-aspartate (*top*) and on total intracellular aspartate levels (*bottom*) in U937 AML cells after 6 hours of incubation. Every dot represents a different clone measured in biological duplicate. **D.** Viability of U937 cells with or without *SLC1A3* knockout, as measured by flow cytometry. Every bar represents a different clone, and every dot a biological replicate. **E.** Cumulative population doublings of mouse MLL-AF9 AML cells cultured at 21%, 2% or 1% oxygen tension in the absence or presence of aspartate in the culture medium at the indicated concentrations. **F.** Viability of AML cells treated with vehicle or TFB-TBOA (250 µM) in the absence or presence of induction chemotherapy (iCT) drugs (10 nM cytarabine and 30 nM doxorubicin) for 24 hours, as measured by flow cytometry. **G.** Viability of U937 cells with or without *SLC1A3* knockout in the absence or presence of iCT drugs (1 µM cytarabine and 30 nM doxorubicin) for 24 hours, as measured by flow cytometry. Every dot represents a different clone measured in biological triplicate. **H.** Viability of MA9−2 AML cells treated with vehicle or iCT drugs (10 nM cytarabine and 30 nM doxorubicin) in the absence or presence (150 µM) of aspartate in the culture medium for 24 hours, as measured by flow cytometry. Data are presented as mean±SD. *P<0.05; ****P<0.0001.

thus indirectly confirming the effect of the compounds *in vivo* (Fig 5A and 5B). In contrast, when we measured the levels in the peripheral blood, no changes were observed (Fig 5A).

We then engrafted non-irradiated C57Bl/6 recipient mice with MA9–2 GFP- and luciferase-expressing AML cells and treated them with TFB-TBOA, UCPH-101 or their respective vehicle starting at day 4 post-engraftment. Tracking of leukemic burden by bioluminescence imaging demonstrated that AML cells were able to grow normally in the presence of either of the aspartate transporter inhibitors, although TFB-TBOA did appear to slightly reduce AML cell growth (Fig 5C and 5D). To further investigate this, we sacrificed mice at day 10, after 6 days of TFB-TBOA treatment, and analyzed the number of GFP+ AML cells in the bone marrow, which confirmed a modest but significant reduction in leukemic burden (Fig 5E).

Next, we combined treatment of AML-engrafted mice with EAAT1 inhibitors (starting at day 7) and iCT (doxorubicin and cytarabine given in a 5+3 regimen, starting at day 9). While iCT prolonged mouse survival, neither TFB-TBOA nor UCPH-101 further enhanced this response (Fig 5F and 5G), thus showing that inhibition of aspartate transporters does not impact the response of AML cells to chemotherapy *in vivo*.

## Discussion

In the current study, we used pharmacological and genetic targeting, metabolic profiling, functional analyses and *in vitro* and *in vivo* AML models to demonstrate that EAAT aspartate/glutamate transporters are not a critical metabolic dependency in AML. Despite its expression in AML cells and upregulation with chemotherapy treatment, EAAT1 inhibition fails to significantly impair leukemic cell viability or enhance the efficacy of standard chemotherapy.

Based on our previous observations that the bone marrow plasma contains an elevated concentration of aspartate and that residual AML cells persisting after iCT highly depend on pyrimidine synthesis [19], which requires aspartate as a substrate, we hypothesized that AML cells rely on aspartate/glutamate transporters of the EAAT family for their growth and survival. We found that AML cells across genetic backgrounds express EAAT1, with the highest expression seen in M4 and M5 AML subtypes, but detected only very low levels of the other EAATs. *In vitro*, pharmacological inhibition of EAAT1 with UCPH-101 reduced intracellular aspartate levels and AML cell viability, but only when the inhibitor was used at 20 µM or higher. In contrast, concentrations of 1–1.5 µM have been shown to fully block aspartate or glutamate uptake in other cell models including HEK293 cells and astrocytes [36–38]. In line with this, we found that the cell death and metabolic changes induced by 20 µM UCPH-101 were aspartate-independent and thus likely the consequence of off-target effects of the inhibitor. Aspartate uptake experiments further revealed that specific EAAT1 inhibition or knockout failed to impact aspartate uptake in AML cells, while the competitive inhibitor TFB-TBOA did have an effect. This shows that AML cells do not depend on EAAT1 for aspartate uptake and may compensate with other EAATs or possess alternative aspartate transport mechanisms. Nevertheless, even though AML cells take up aspartate from the culture medium, they do not need this source to grow in culture or survive chemotherapy.

Given that low oxygen tensions limit the ability of cancer cells to synthetize aspartate [22], exogenous aspartate may become more important as a nutrient under hypoxic conditions such as those found in the bone marrow [23]. However, our findings suggest that even under hypoxia, AML cells exhibit a high degree of resilience to aspartate deprivation.

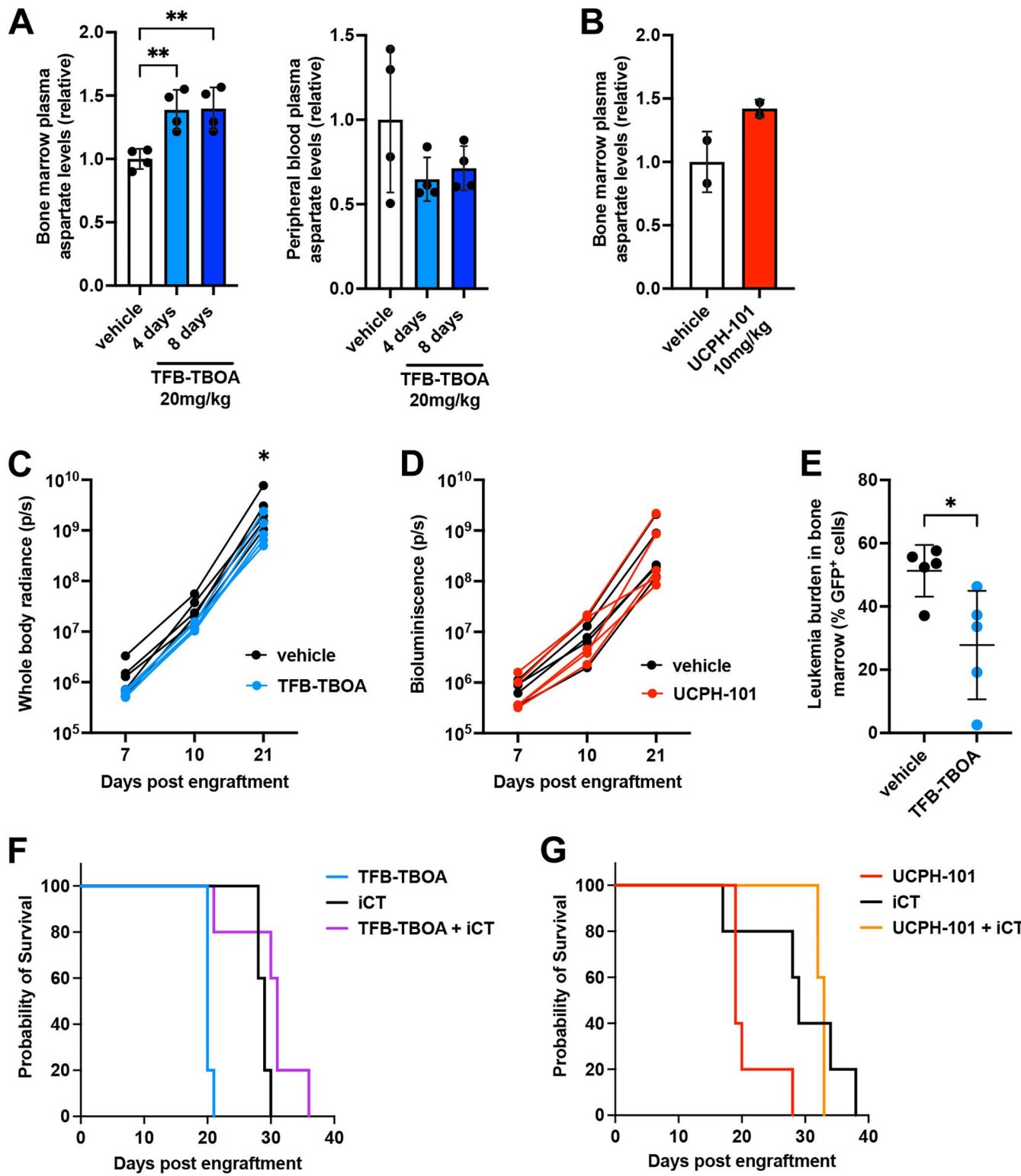

**Fig 5. Inhibition of EAAT1 does not affect AML cells *in vivo*. A.** Relative aspartate levels in the bone marrow plasma or peripheral blood plasma of mice treated with vehicle for 8 days or 20 mg/kg TFB-TBOA for 4 or 8 days, as measured by GC-MS. **B.** Relative aspartate levels in the bone marrow plasma of mice treated with vehicle or 10 mg/kg UCPH-101 for 2 days, as measured by GC-MS. **C-D.** Leukemic burden as measured by whole body

radiance of mice engrafted with MA9-2 GFP- and luciferase-expressing AML cells and treated daily with vehicle, 20 mg/kg TFB-TBOA (C) or 10 mg/kg UCPH-101 (D) starting at day 4 post engraftment. **E.** Leukemic burden in the bone marrow at day 10, as measured by flow cytometry, of mice engrafted with MA9-2 GFP- and luciferase-expressing AML cells and treated daily with vehicle or 20 mg/kg TFB-TBOA starting at day 4 post engraftment. **F-G.** Kaplan-Meier survival curves of mice engrafted with MA9-2 AML cells and treated daily with vehicle, 20 mg/kg TFB-TBOA (F) or 10 mg/kg UCPH-101 (G) starting at day 7 post engraftment, with or without concomitant induction chemotherapy (iCT; doxorubicin 3 mg/kg and cytarabine 100 mg/kg given in a 5+3 regimen) treatment started at day 9 post engraftment. Data are presented as mean±SD. *$P < 0.05$; **$P < 0.01$.

Similarly, we found that the inhibition of EAATs had only mild effects on AML growth or response to chemotherapy *in vivo*. These findings underscore the intrinsic metabolic plasticity of leukemic blasts, which, like their normal hematopoietic counterparts, are adept at surviving and proliferating in the oxygen-poor bone marrow microenvironment. Moreover, a recent study indicates that cancer cells rapidly adapt to a reduction in intracellular aspartate levels by establishing a new steady-state and that the negative effects of electron transport chain inhibition only manifest when aspartate levels fall below a critical threshold [39]. In addition to EAATs, AML cells may use other transporters to take up aspartate and could additionally replenish their aspartate pools not only by synthesizing aspartate via glutamate-oxaloacetate transaminases, but also by converting asparagine to aspartate, recycling aspartate from intracellular protein turnover, importing extracellular proteins via macropinocytosis or engaging in metabolic exchange through cell-cell interactions [20]. Such metabolic plasticity would allow AML cells to circumvent blockade of a single pathway, further illustrating the challenge of targeting individual metabolic nodes in malignancy.

The lack of effect of EAAT1 inhibition in AML stands in contrast to a previous report which identified this transporter as a metabolic vulnerability in T-cell acute lymphoblastic leukemia [40]. However, this study implicated EAAT1 as a mitochondrial glutamate/aspartate transporter involved in glutamine-to-aspartate conversion rather than as a cell surface transporter involved in aspartate uptake, and used UCPH-101 at 25 µM without validating whether its effects were due to aspartate deprivation. A direct comparison of their results to ours is therefore not evident.

Taken together, our findings suggest that while extracellular aspartate and its transport via EAATs may contribute to aspartate pools in AML, these transporters do not constitute a targetable vulnerability for this cancer type. Future therapeutic strategies may need to involve the simultaneous disruption of several aspartate acquisition routes to overcome metabolic redundancy. However, such approaches must carefully consider toxicity to normal hematopoietic progenitors, which may share these adaptive traits. Continued efforts to identify selective vulnerabilities in AML metabolism, possibly through synthetic lethality or metabolic bottlenecks unique to malignant cells, will be essential for advancing more effective therapies.

## Supporting information

**S1 Fig.** A-B. Expression of *SLC1A1* (A) and *SLC1A2* (B) in AML cells across different patient genetic subgroups in the BEAT-AML cohort. C-E. Expression of *SLC1A1* (**C**), *SLC1A2* (**D**) and *SLC1A3* (**E**) in bone marrow cells of patients with different hematological cancers or healthy donors from the MILE study cohort. **F.** EAAT1 protein levels in bone marrow cells of AML patients stratified according to FAB classification in the TCGA cohort.
(TIF)

**S2 Fig.** A. Expression of *SLC1A3* in U937 cells after nucleofection with CAS9 protein and sgRNAs targeting *SLC1A3* or a control sgRNA. Every dot represents a different clone measured in biological duplicate. **B.** Cell cycle analysis of U937 cells with or without *SLC1A3* knockout as measured by flow cytometry. **C.** Viability of U937 cells with or without *SLC1A3* knockout in the absence or presence of UCPH-101 (20 µM) for 24 hours, as measured by flow cytometry. Every dot represents a different clone measured in biological triplicate. Data are presented as mean±SD. ****$P < 0.0001$.
(TIF)

**S1 Raw Data. Excel file containing all raw data used for the generation of the graphs included in this manuscript.**
(XLSX)

## Acknowledgments

We thank Isabelle Gerin, Francesco Caligiore and Guido Bommer from the Biochemistry and Metabolic Research Group at the de Duve Institute (UCLouvain, Belgium) for access to and help with GC-MS analysis. We also thank Nicolas Dauguet from the CYTF platform at the de Duve Institute for help with flow cytometry analysis, and all members of the Cellular Metabolism and Microenvironment Laboratory for helpful discussions.

## Author contributions

**Conceptualization:** Nick van Gastel.

**Data curation:** Hernán A. Tirado, Jean Jacobs, Nick van Gastel.

**Formal analysis:** Hernán A. Tirado, Nithya Balasundaram, Jean Jacobs, Nick van Gastel.

**Funding acquisition:** Nick van Gastel.

**Investigation:** Hernán A. Tirado, Nithya Balasundaram, Fleur Leguay, Lotfi Laaouimir.

**Methodology:** Hernán A. Tirado, Nithya Balasundaram, Jean Jacobs, Fleur Leguay, Lotfi Laaouimir.

**Project administration:** Nick van Gastel.

**Supervision:** Nick van Gastel.

**Validation:** Nick van Gastel.

**Visualization:** Hernán A. Tirado, Nick van Gastel.

**Writing – original draft:** Hernán A. Tirado, Nick van Gastel.

**Writing – review & editing:** Nick van Gastel.

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
