## [Decision Letter · Decision Letter 0]

19 Aug 2025

Dear Dr. van Gastel,

Thank you for submitting your manuscript to PLOS ONE. After careful consideration, we feel that it has merit but does not fully meet PLOS ONE’s publication criteria as it currently stands. Therefore, we invite you to submit a revised version of the manuscript that addresses the points raised during the review process.

We look forward to receiving your revised manuscript.

Kind regards,

Mohamed Abdelkarim

Academic Editor

PLOS ONE

Journal Requirements:

2. Thank you for stating the following financial disclosure: [This work was supported by the Belgian Foundation Against Cancer [F/2020/1440, F/2024/2556], the Fund for Scientific Research - FNRS [F.R.S.-FNRS; A5/5-CQ/135, M4/1/2/5-MIS/BEJ] and the de Duve Institute (N.v.G.). H.A.T. was supported by a doctoral research fellowship from the Fund for Research Training in Industry and Agriculture (FRIA) [1.E.027.22] and by a Bourse du Patrimoine from the UCLouvain. L.L. was supported by a FRIA doctoral research fellowship [1.E017.24], N.B. was supported by a post-doctoral Chargée de Recherches fellowship from the F.R.S.-FNRS [1.B.027.24].].

3. We note that there is identifying data in the Supporting Information file < Tirado et al_EAAT1 in AML_Supporting Information.pdf  >. Due to the inclusion of these potentially identifying data, we have removed this file from your file inventory. Prior to sharing human research participant data, authors should consult with an ethics committee to ensure data are shared in accordance with participant consent and all applicable local laws.

-Location data

Please remove or anonymize all personal information (Name), ensure that the data shared are in accordance with participant consent, and re-upload a fully anonymized data set. Please note that spreadsheet columns with personal information must be removed and not hidden as all hidden columns will appear in the published file.

4. Please include captions for your Supporting Information files at the end of your manuscript, and update any in-text citations to match accordingly. Please see our Supporting Information guidelines for more information: http://journals.plos.org/plosone/s/supporting-information .

Reviewers' comments:

Reviewer's Responses to Questions

**Comments to the Author**

1. Is the manuscript technically sound, and do the data support the conclusions?

Reviewer #1: Partly

Reviewer #2: Partly

2. Has the statistical analysis been performed appropriately and rigorously?

Reviewer #1: Yes

Reviewer #2: Yes

3. Have the authors made all data underlying the findings in their manuscript fully available?

Reviewer #1: Yes

Reviewer #2: Yes

4. Is the manuscript presented in an intelligible fashion and written in standard English?

Reviewer #1: Yes

Reviewer #2: Yes

Reviewer #1: This manuscript investigates the role of the EAAT1 aspartate/glutamate transporter 1 in acute myeloid leukemia (AML). According to the fact that AML's dependency on pyrimidine biosynthesis and aspartate is an essential substrate for this process, the author first hypothesized that targeting EAAT1-mediated aspartate uptake could serve as a therapeutic strategy. In this study, they demonstrate that EAAT1 is broadly expressed in AML cells, particularly enriched in M4 and M5 subtypes, with increased expression following chemotherapy. In vitro, pharmacological inhibition of EAAT1 reduces AML cell viability, but this effect is independent of aspartate levels, suggesting off-target effects. In vivo experiments show minimal impact of EAAT1 inhibition on AML growth and chemotherapy sensitivity. Overall, this study concludes that despite EAAT1 expression in AML, it does not constitute a targetable metabolic vulnerability due to metabolic plasticity and multiple compensatory mechanisms.

Major suggestions:

1. For evaluating the effect of EAAT1 inhibitor UCPH-101 and the measurement of aspartate uptake, using direct radio-labeled or fluorescent-labeled aspartate uptake assays rather than relying solely on indirect metabolomics evidence would be better.

2. Regarding to the observed metabolic changes induced by the UCPH-101, can it be restored by supplementation of aspartate to the cells? And what pathways are those changed metabolites related to?

3. While the off-target effects of UCPH-101 are mentioned, the specific mechanism of action remains unclear. The proteomics or drug target screening experiments could be performed to identify the actual targets of UCPH-101 at 20μM concentration.

4. To investigate the biological function of EAAT1 in AML cells, some gene editing methods can be used to study its function, such as shRNA and CRISPR/Cas9 based knockout of EAAT1 would provide more specific functional validation.

5. The manuscript mentions multiple potential aspartate compensation pathways but lacks more detailed validation, maybe glutamate-oxaloacetate transaminase activity, asparaginase expression, and micropinocytosis can be examined as compensatory mechanisms.

Overall, this is a well-designed negative result study with important value for understanding AML metabolism. I recommend publication after addressing the above experimental suggestions to provide more comprehensive scientific evidence.

Reviewer #2: Manuscript ID: PONE-D-25-37118

Manuscript title: The EAAT1 aspartate/glutamate transporter is dispensable for acute myeloid leukemia cell growth and response to therapy

Authors aimed to identify the molecular mechanisms of metabolic regulation related to cell growth and therapy resistance of acute myeloid leukemia. Inspired by the public datasets/databases supporting that ETTA1 (SLC1A3) is upregulated in AML patients, authors hypothesized that the cellular uptake of aspartate is a key factor in the AML development and treatment. Two inhibitors were utilized in in vitro and in vivo investigations, and rescue experiments (addition of metabolites or aspartate) were conducted aiming to uncover the mechanisms of observed phenotypes. The inhibitor UCPH-101 exhibited more potent phenotypes compared to TFB-TBOA in reducing cellular metabolic rates, viability and division. Assisted by the metabolic profiling detected by mass spectrometry, evidence was presented that UCPH-101 inhibitor caused the disorder of metabolism in vitro. Rescue attempts, though, by means of metabolite addition, failed to reverse the inhibitor phenotypes in vitro. Extracellular aspartate treatment or inhibitors did not seem to affect the sensitivity of AML cells to in vitro or in vivo chemotherapy.

My major concern:

Authors reached the conclusion that EAAT1 is dispensable for the cell growth and therapy response of AML by performing their experiments with inhibitors. In the discussion, authors acknowledged a common problem with inhibitor use, which is that the off-target effect can not be excluded. To reach that conclusion of authors, specific knockdown of EAAT1 will be needed as a treatment for most experiments. However, presented results are sufficient to conclude that extracellular aspartate is dispensable for AML cell growth and response to chemotherapy.

Minor concerns:

1. Line 265, all tested cell lines in Fig. 2B need to be listed in text.

2. Line 281-283, more details of text description may help readers instead of a summary.

3. Line 283, Fig. 3B aspartate is not an essential amino acid.

4. Fig. 3A and 3B, it looks that Fig. 3B is another presentation of aspartate level from Fig. 3A.

5. Line 294, Fig. 3D-J? Maybe a typo?

6. Line 300-301, it is confusing here. In Fig. 2A, UCPH-101 showed cytotoxicity at 10 µM. Plus, this description cannot be supported by Fig. 4A.

7. Line 334-335, the increase of aspartate level cannot support the idea of reduced cell consumption or validate the inhibitor effect of aspartate uptake.

8. Line 355-356, this description is confusing as inhibitors did affect metabolism shown in Fig. 3A.

9. Line 357-358, this description is confusing as inhibitors did affect the viability of AML cells in Fig. 2C.

10. Line 580, Panel D might be a typo?

**Do you want your identity to be public for this peer review?** For information about this choice, including consent withdrawal, please see our Privacy Policy

Reviewer #1: No

Reviewer #2: No

---

## [Author Response · Author response to Decision Letter 1]

17 Dec 2025

Response to Reviewer Comments

-----

Reviewer #1: This manuscript investigates the role of the EAAT1 aspartate/glutamate transporter 1 in acute myeloid leukemia (AML). According to the fact that AML's dependency on pyrimidine biosynthesis and aspartate is an essential substrate for this process, the author first hypothesized that targeting EAAT1-mediated aspartate uptake could serve as a therapeutic strategy. In this study, they demonstrate that EAAT1 is broadly expressed in AML cells, particularly enriched in M4 and M5 subtypes, with increased expression following chemotherapy. In vitro, pharmacological inhibition of EAAT1 reduces AML cell viability, but this effect is independent of aspartate levels, suggesting off-target effects. In vivo experiments show minimal impact of EAAT1 inhibition on AML growth and chemotherapy sensitivity. Overall, this study concludes that despite EAAT1 expression in AML, it does not constitute a targetable metabolic vulnerability due to metabolic plasticity and multiple compensatory mechanisms.

Overall, this is a well-designed negative result study with important value for understanding AML metabolism. I recommend publication after addressing the above experimental suggestions to provide more comprehensive scientific evidence.

A: We thank the reviewer for their critical assessment of our manuscript, overall positive evaluation and helpful suggestions. We also appreciate that the reviewer mentions that our study is well-designed and has important value for the field. Please find below our detailed response to the major suggestions.

Major suggestions:

1. For evaluating the effect of EAAT1 inhibitor UCPH-101 and the measurement of aspartate uptake, using direct radio-labeled or fluorescent-labeled aspartate uptake assays rather than relying solely on indirect metabolomics evidence would be better.

A: Thank you for this comment, and we fully agree that a robust aspartate uptake assay was lacking to allow for correct interpretation of the effects of the inhibitors. Given that a fluorescent-labeled aspartate currently does not exist and we do not have experience with radio-labeled aspartate, we decided to perform aspartate uptake assays with stable isotope 13C-labeled aspartate which can be measured by GC-MS. These data are now included in Figure 4A-B, showing that while UCPH-101 does not impact aspartate uptake (even though it reduces cellular aspartate levels), TFB-TBOA does reduce aspartate uptake by AML cells at the concentration used in our experiments. The description of these data is now included on line 333-342 of the revised manuscript, and in the discussion on line 412-427.

2. Regarding to the observed metabolic changes induced by the UCPH-101, can it be restored by supplementation of aspartate to the cells? And what pathways are those changed metabolites related to?

A: We agree that the effects of UCPH-101 are quite puzzling. While the inhibitor induces AML cell death in vitro at higher concentrations, it does not block aspartate uptake. However, it does lead to extensive metabolic effects including reduction of the levels of aspartate and TCA cycle metabolites, while increasing the levels of essential amino acids such as leucine, isoleucine, lysine, methionine, phenylalanine and valine. We have performed further metabolomic analysis by GC-MS of AML cells treated with UCPH-101 with or without dimethyl-aspartate rescues (Figure 3D), but apart from the normalization of intracellular aspartate levels, none of the other metabolic changes were restored by aspartate supplementation. This indicates that UCPH-101 impacts AML cell metabolism independent of its effects on intracellular aspartate levels. The description of these data is now included on line 320-323 of the revised manuscript

3. While the off-target effects of UCPH-101 are mentioned, the specific mechanism of action remains unclear. The proteomics or drug target screening experiments could be performed to identify the actual targets of UCPH-101 at 20μM concentration.

A: Thank you for this comment. We agree that we currently do not have any idea of how UCPH-101 induces AML cell death in vitro when used at higher concentrations. However, given that it would likely take a large amount of effort, money and time to figure this out, and given the lack of effects of UCPH-101 on AML cells in vivo (at concentrations shown previously to be effective in other models), we do not find it pertinent or useful at this time to try and identify the actual target(s) of UCPH-101 in AML cells.

4. To investigate the biological function of EAAT1 in AML cells, some gene editing methods can be used to study its function, such as shRNA and CRISPR/Cas9 based knockout of EAAT1 would provide more specific functional validation.

A: We fully agree with this comment, which was also raised by reviewer 2, and have therefore created U937 cell clones lacking EAAT1 using a CRISPR/Cas9 approach (with two different sgRNAs targeting SLC1A3 that encodes for EAAT1). In line with our other findings, these cells did not show a reduction of aspartate uptake, viability, proliferation or changes in their response to chemotherapy, further confirming that EAAT1 is dispensable for aspartate uptake by AML cells, their growth or response to therapy. These data are now included in Figure 4, panels C, D and G as well as in Supplementary Figure S2. In the text, these results are discussed on lines 344-352 and 366-372.

When generating the knockout cell clones, we also noticed that even though we had a good knockout efficiency with strong reduction of SLC1A3 mRNA levels, there were no changes detected in EAAT1 by flow cytometry. This made us question the specificity of our EAAT1 antibody, and we therefore decided to replace the flow cytometry data in the original Figure 1 with gene expression data for SLC1A3 in human AML cells (new Figure 1D) and for Slc1a1, Slc1a2 and Slc1a3 in mouse AML cells (new Figure 1E). The interpretations however did not change, as we still detected SLC1A3 expression across multiple human AML cell lines, and higher expression of Slc1a3 compared to the other two isoforms in the mouse AML cell line.

5. The manuscript mentions multiple potential aspartate compensation pathways but lacks more detailed validation, maybe glutamate-oxaloacetate transaminase activity, asparaginase expression, and micropinocytosis can be examined as compensatory mechanisms.

A: Thank you for this comment. We indeed discuss multiple potential compensation mechanisms in the Discussion section of the paper. However, the focus of the current study was on the role of the EAAT transporters, which we found to not be required for maintaining aspartate levels or viability in AML cells. Given the key role of aspartate in cellular metabolism, AML cells must therefore use other mechanisms to maintain their aspartate levels under different conditions, but this will be the focus of future studies of our group. We therefore do not think that a more detailed validation of the role of glutamate-oxaloacetate transaminases, asparaginase or macropinocytosis in maintaining AML cell aspartate levels falls under the scope of the current study focused on EAATs.

Reviewer #2: Manuscript ID: PONE-D-25-37118

Manuscript title: The EAAT1 aspartate/glutamate transporter is dispensable for acute myeloid leukemia cell growth and response to therapy

Authors aimed to identify the molecular mechanisms of metabolic regulation related to cell growth and therapy resistance of acute myeloid leukemia. Inspired by the public datasets/databases supporting that ETTA1 (SLC1A3) is upregulated in AML patients, authors hypothesized that the cellular uptake of aspartate is a key factor in the AML development and treatment. Two inhibitors were utilized in in vitro and in vivo investigations, and rescue experiments (addition of metabolites or aspartate) were conducted aiming to uncover the mechanisms of observed phenotypes. The inhibitor UCPH-101 exhibited more potent phenotypes compared to TFB-TBOA in reducing cellular metabolic rates, viability and division. Assisted by the metabolic profiling detected by mass spectrometry, evidence was presented that UCPH-101 inhibitor caused the disorder of metabolism in vitro. Rescue attempts, though, by means of metabolite addition, failed to reverse the inhibitor phenotypes in vitro. Extracellular aspartate treatment or inhibitors did not seem to affect the sensitivity of AML cells to in vitro or in vivo chemotherapy.

A: We thank the reviewer for their critical assessment of our manuscript, overall positive evaluation and helpful suggestions. Please find below our detailed response to the major and minor concerns.

My major concern:

Authors reached the conclusion that EAAT1 is dispensable for the cell growth and therapy response of AML by performing their experiments with inhibitors. In the discussion, authors acknowledged a common problem with inhibitor use, which is that the off-target effect can not be excluded. To reach that conclusion of authors, specific knockdown of EAAT1 will be needed as a treatment for most experiments. However, presented results are sufficient to conclude that extracellular aspartate is dispensable for AML cell growth and response to chemotherapy.

A: We fully agree with this comment, which was also raised by reviewer 1, and have therefore created U937 cell clones lacking EAAT1 using a CRISPR/Cas9 approach (with two different sgRNAs targeting SLC1A3 that encodes for EAAT1). In line with our other findings, these cells did not show a reduction of aspartate uptake, viability, proliferation or changes in their response to chemotherapy, further confirming that EAAT1 is dispensable for aspartate uptake by AML cells, their growth or response to therapy. These data are now included in Figure 4, panels C, D and G as well as in Supplementary Figure S2. In the text, these results are discussed on lines 344-352 and 366-372.

When generating the knockout cell clones, we also noticed that even though we had a good knockout efficiency with strong reduction of SLC1A3 mRNA levels, there were no changes detected in EAAT1 by flow cytometry. This made us question the specificity of our EAAT1 antibody, and we therefore decided to replace the flow cytometry data in the original Figure 1 with gene expression data for SLC1A3 in human AML cells (new Figure 1D) and for Slc1a1, Slc1a2 and Slc1a3 in mouse AML cells (new Figure 1E). The interpretations however did not change, as we still detected SLC1A3 expression across multiple human AML cell lines, and higher expression of Slc1a3 compared to the other two isoforms in the mouse AML cell line.

Minor concerns:

1. Line 265, all tested cell lines in Fig. 2B need to be listed in text.

A: We have listed all the cell lines that were tested in the text (line 289-290 in revised manuscript)

2. Line 281-283, more details of text description may help readers instead of a summary.

A: Thank you for this remark, we have added a more detailed explanation of the observed changes on lines 309-317, which we hope will help the readers in understanding the implications of our findings.

3. Line 283, Fig. 3B aspartate is not an essential amino acid.

A: We agree, and aspartate was not the essential amino acid(s) we refer to in line 283 (now line 309). However, our lack of detail in explaining the GC-MS results may have caused some difficulty in following our reasoning, which we hope to have better addressed now, as explained in the comment above.

4. Fig. 3A and 3B, it looks that Fig. 3B is another presentation of aspartate level from Fig. 3A.

A: This is indeed the case. We added aspartate in the separate graph in Figure 3B to allow an easier interpretation of the results of this metabolite and add statistical information given its central role in our study. Including separate graphs for all metabolites would create a large number of graphs that would not provide that much more insight compared to the heat map. We have detailed in the figure legend that Figure 3B is another presentation of the aspartate data from 3A, and have done the same for the new Figures 3C and 3D.

5. Line 294, Fig. 3D-J? Maybe a typo?

A: This was indeed a typo and has been corrected.

6. Line 300-301, it is confusing here. In Fig. 2A, UCPH-101 showed cytotoxicity at 10 µM. Plus, this description cannot be supported by Fig. 4A.

A: We agree that this part was confusing, given that we showed that UCPH-101 has off-target effects but then still used it at a lower concentration (that did not induce cell death but did have an impact on overall cell activity as revealed by the MTT assay in Figure 2A) in Figure 4 to examine synergy with low oxygen tensions or chemotherapy. We have therefore decided to remove the data with UCPH-101 at 10 µM from Figure 4 (original panels A and C). Figure 4 has now been updated to instead include data on the effects of the inhibitors on aspartate uptake (panels A & B) and the studies we did with SLC1A3-knockout U937 cells (panels C, D and G).

7. Line 334-335, the increase of aspartate level cannot support the idea of reduced cell consumption or validate the inhibitor effect of aspartate uptake.

A: We agree that this is very indirect evidence of the effect of the inhibitors. However, given that treatment with the inhibitors likely does not impact intracellular aspartate levels in AML cells (cfr. our in vitro data) and that in vivo stable isotope tracing experiments are very complex and expensive, we use the observed increase in aspartate levels in the bone marrow plasma as a proxy that our inhibitors are reaching the bone marrow and blocking uptake of aspartate by some cells in the bone marrow. To make it more clear to the reader that this is indirect evidence, we have modified our wording (now in line 385).

8. Line 355-356, this description is confusing as inhibitors did affect metabolism shown in Fig. 3A.

A: With the new aspartate uptake and SLC1A3 knockout data we believe that we now have sufficient evidence to conclude that EAAT transporters are not a critical metabolic dependency in AML. We agree that UCPH-101 at higher doses has strong effects on AML metabolism, but it is now clear that this is not due to its inhibition of EAAT1 or of aspartate uptake by AML cells.

9. Line 357-358, this description is confusing as inhibitors did affect the viability of AML cells in Fig. 2C.

A: As mentioned in the response to the comment above, we believe we have sufficient evidence now to conclude that the effects of UCPH-101 on AML cell viability are independent of EAAT1 inhibition.

10. Line 580, Panel D might be a typo?

A: This was indeed a typo and has been corrected.

Response to Editorial Comments

-----

Identifying data in the Supporting Information:

In the decision e-mail, it is noted that there is identifying data in the Supporting Information file which has therefore been removed from the file inventory. However, it is not clear to me which identifying data is referred to, since the Supporting Information file does not contain any names, initials, addresses, ages, dates, contact information, location data or ID numbers. Supplementary Figure S1, panel F does contain some subgroups with only a few individuals, but there are no indirect identifiers included that could lead to identification in any way. Furthermore, all the data included in the Supporting Information Figure 1 are publicly available data from prior studies on AML patients showing expression or protein levels of SLC1A1/SLC1A2/SLC1A3. These data were not generated by us and are available to the public via www.bloodspot.eu, www.vizome.org or proteomics.leylab.org as we describe in the Methods section.

Updated Financial Disclosure and Amended ‘Role of Funder’ statement:

The funding section has been updated to include fundings for the additional author, correct some award numbers that were incorrect, and since the funders had no role in this study, we have also added this information. Financial Disclosure section can therefore be amended to:

“This work was supported by the Belgian Foundation Against Cancer [F/2020/1440, F/2024/2556], the Fund for Scientific Research - FNRS [F.R.S.-FNRS; A5/5-CQ/135, M4/1/2/5-MIS/BEJ] and the de Duve Institute. H.A.T. and J.J. we

---

## [Decision Letter · Decision Letter 1]

26 Jan 2026

The EAAT1 aspartate/glutamate transporter is dispensable for acute myeloid leukemia cell growth and response to therapy

PONE-D-25-37118R1

Dear Dr. Nick van Gastel,

We’re pleased to inform you that your manuscript has been judged scientifically suitable for publication and will be formally accepted for publication once it meets all outstanding technical requirements.

Kind regards,

Mohamed Abdelkarim

Academic Editor

PLOS One

Additional Editor Comments (optional):

Reviewers' comments:

Reviewer's Responses to Questions

**Comments to the Author**

Reviewer #1: All comments have been addressed

Reviewer #2: All comments have been addressed

2. Is the manuscript technically sound, and do the data support the conclusions?

Reviewer #1: Yes

Reviewer #2: (No Response)

3. Has the statistical analysis been performed appropriately and rigorously?

Reviewer #1: Yes

Reviewer #2: (No Response)

4. Have the authors made all data underlying the findings in their manuscript fully available?

Reviewer #1: No

Reviewer #2: (No Response)

5. Is the manuscript presented in an intelligible fashion and written in standard English?

Reviewer #1: Yes

Reviewer #2: (No Response)

Reviewer #1: I have reviewed the authors' response and the revised manuscript. The revisions adequately address the primary concerns raised by the reviewers. The addition of the 13C-labeled aspartate uptake assay and the CRISPR/Cas9 knockout data effectively supports the conclusion that EAAT1 is not a viable therapeutic target in AML, while clarifying the specificity issues regarding the inhibitors.

Regarding the points not addressed experimentally, such as identifying the specific off-targets of UCPH-101 or detailed compensation mechanisms, the authors' rebuttals are logical and consistent with the scope of the current study. The transparent correction of the antibody data further ensures the accuracy of the findings. Overall, the modifications are sufficient, and I believe the manuscript is now in an acceptable state.

Reviewer #2: (No Response)

**Do you want your identity to be public for this peer review?** For information about this choice, including consent withdrawal, please see our Privacy Policy

Reviewer #1: No

Reviewer #2: No

---

## [Editor Report · Acceptance letter]

PONE-D-25-37118R1

PLOS One

Dear Dr. van Gastel,

I'm pleased to inform you that your manuscript has been deemed suitable for publication in PLOS One. Congratulations! Your manuscript is now being handed over to our production team.

Kind regards,

on behalf of

Dr. Mohamed Abdelkarim

Academic Editor

PLOS One